# Investigation of Glycine Polymorphic Transformation by In Situ ATR-FTIR and FT-Raman Spectroscopy

**Zunhua Li** [1,*,†] **and Bowen Zhang** [2,†]

1 College of Chemistry and Bioengineering, Hunan University of Science and Engineering, Yongzhou 425199, China
2 Department of Chemical Engineering, Kyung Hee University, Yongin 17104, Korea
* Correspondence: zhli@khu.ac.kr; Tel.: +86-15874681775; Fax: +86-746-6381164
† These authors contributed equally to this work.

**Abstract:** The solution-mediated phase transformation of α-form to γ-form glycine, including dissolution of metastable α-form, nucleation, and growth of stable γ-form during polymorphic transformation, was investigated using in situ attenuated total-reflectance Fourier transform infrared spectroscopy (ATR-FTIR) and Fourier transform Raman spectroscopy (FT-Raman). The mechanistic influence of operating parameters such as agitation speed, crystallization temperature, α-form seed concentration, and NaCl concentration on polymorphic phase transformation was examined. When the agitation speed, crystallization temperature, and NaCl concentration were increased, the polymorphic transformation process was improved due to the promotion of nucleation and growth of stable γ-form, in addition to the promotion of dissolution of metastable α-form. Moreover, the time to induce γ-form nucleation and complete conversion of α-form to γ-form was also reduced with increasing α-form seed concentration.

**Keywords:** phase transformation; glycine; in situ ATR-FTIR; FT-Raman spectroscopy





## 1. Introduction

Polymorphism refers to the various crystal forms resulting from the crystallization of the same substance. Owing to the various arrangements of crystals, polymorphs have varying properties, such as crystal habit, density, solubility, hardness, compatibility, bioavailability, chemical stability, etc. [1–3]. Crystallization is an indispensable unit operation in the production and purification of many drugs and chemicals. Therefore, the study of polymorphism and polymorphism transformation is important in pharmacology and the fine chemical industry [4,5].

Polymorphic crystallization can be affected by many factors, such as temperature, supersaturation, additives, solvent, hydrodynamic conditions, and seeding [6–11]. According to Ostwald's rule of stages, polymorphic crystallization is theoretically easy to first form a metastable form with high free energy because there is a low energy barrier to overcome [12]. Afterwards, the metastable phase can undergo phase transformation to form a more stable form. The stable form is often favored when considering the safety and stability of the drug during storage. However, it is worth noting that the stable form usually has the lowest solubility, which may limit the bioavailability of drugs. A typical example is the antiviral drug ritonavir, the metastable form I of ritonavir used to treat acquired immunodeficiency syndrome (AIDS). However, when form I was converted into the more stable form II, the solubility was considerably reduced so that the therapeutic effect could not be achieved [13]. Owing to the above facts, the polymorphic phase transformation during polymorph crystallization has received extensive attention. For instance, Chan et al. studied the polymorphic transformation of 32 drugs under mechanical treatment [14]. Pirttimäki et al. reported the polymorphic transformation of caffeine during tableting and found that the degree of phase transformation increased with increasing

compression time and compression pressure [15]. According to Lin et al., grinding induced the polymorphic transformation of famotidine form B to form A [16]. Solvent-mediated phase transformation is also a frequent phenomenon in polymorphic crystallization. It has been reported that factors such as additives, solvents, seeds, temperature, etc., can influence solvent-mediated phase transitions by affecting the nucleation and growth of stable forms [17–20]. Kobari et al. predicted that phase transformation would be accelerated when increasing the secondary nucleation rate of the stable form using a mathematical model [21].

However, as reported by Threlfall, polymorphic crystallization is a complex behavior that may have been severely overlooked, as when they attempted to repeat 20 polymorphic crystallization experiments reported in the literature, more than 50% of the experiments were not reproducible [22]. This means that a more in-depth study and understanding of polymorphic crystallization and the phase transformation processes are necessary. Furthermore, in most previous studies, the characterization objects were solid samples, and the analysis of solutions is relatively rare. Owing to the differences in solubility between polymorphs, the process of phase transformation is also accompanied by a change in solution concentration. Therefore, if the solution concentration can be accurately detected in real time, it can not only help to track the phase transformation process in time but also contribute to improved understanding of the complex phase transformation process.

Glycine is the simplest amino acid and is widely used in pharmaceutical and food industries. Three polymorphs of glycine have been reported under ambient conditions: $\alpha$, $\beta$, and $\gamma$ form, which follow the relative thermodynamic stability order of $\gamma > \alpha > \beta$ [23,24]. Metastable $\alpha$-glycine can form spontaneously from pure aqueous crystallization [25,26]. $\gamma$-glycine is the thermodynamically stable form, but it is more difficult to form in pure aqueous solution than $\alpha$-glycine and can usually be obtained from solutions containing additives such as acids, bases, or salts [25,27,28]. $\beta$-glycine is extremely unstable and can be grown from water–alcohol mixtures, but it rapidly converts to $\alpha$-glycine upon exposure to water [29,30].

In this work, glycine was selected as the model substance, and the polymorphic phase transformation behavior of $\alpha$-form glycine to $\gamma$-form was studied using Fourier transform Raman (FT-Raman) spectroscopy and in situ attenuated total-reflectance Fourier transform infrared (ATR-FTIR) spectroscopy. As such, changes in both solid composition and solution concentration during phase transformation were monitored. Furthermore, influences of operating parameters such as agitation speed, crystallization temperature, seed concentration, and the solution concentration of NaCl on polymorphic phase transformation were examined. This work provides important insights for improved understanding and control of phase transitions.

## 2. Experiment Section

### 2.1. Materials

Glycine was purchased from Sigma-Aldrich (Saint Louis, MO, USA, $\alpha$-form, purity $\geq$ 99%). The $\alpha$-form and $\gamma$-form glycine crystals were obtained following a previously established protocol [18]. NaCl was purchased from Samchun Chem. (Seoul, South Korea, purity $\geq$ 99.5%). Deionized water was obtained with a Millipore ultrapure water system (Applied Membranes Inc., Vista, CA, USA).

### 2.2. Solution-Mediated Transformation Experiments

The experiments were conducted in a mixing tank crystallizer, as shown in Figure 1. A circulator was used for temperature control, ATR-FTIR was installed for in situ monitoring of the glycine concentration, and a thermocouple was used for temperature monitoring. Volumes of 200 mL of sufficiently glycine-saturated solutions (198.9 g/L at 20 °C, 245.7 g/L at 30 °C, and 278.8 g/L at 40 °C) with respect to $\alpha$-form were prepared at a corresponding temperature. A sufficient amount of $\alpha$-form crystals seeds (60 or 120 g/L, 212–300 μm) and a sufficient amount of NaCl (80 or 120 g/L) were added into the prepared saturated solutions,

and the suspension was stirred at a sufficient agitation speed (750 or 1200 rpm). A portion of the suspension was collected at set intervals for offline monitoring of crystallization. During the sampling, the crystallizer was simultaneously refilled with feed solution, and the sample suspension was immediately filtered by a vacuum pump and dried for 24 h in a convection oven at 60 °C.

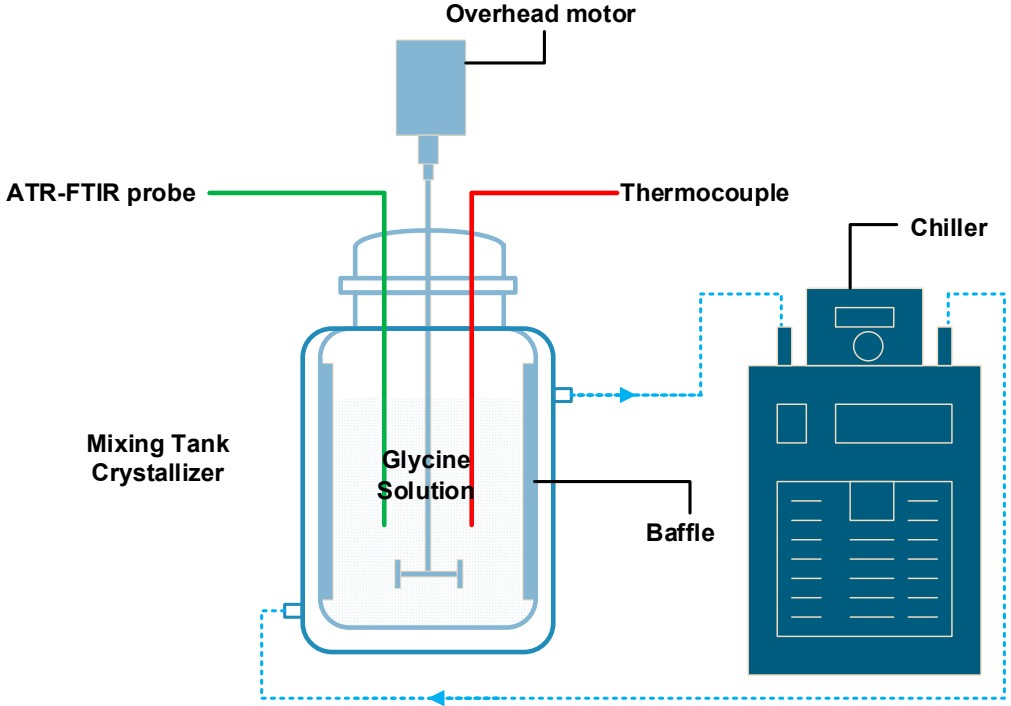

**Figure 1.** Schematic of the experimental setup in the mixing tank crystallizer consisting of in situ ATR-FTIR and a thermocouple probe.

### 2.3. Characterization

Powder X-ray diffraction (PXRD, Bruker D8-ADVANCE, Karlsruhe, Germany, with Cu Kα radiation (wavelength = 1.54 Å)) was used to determine the PXRD patterns of glycine crystals at room temperature. The applied tube voltage and tube current were 45 kV and 15 mA, respectively. Each sample was continuously scanned from 10 to 40° at a constant scan rate of 0.02 °/s. FT-Raman spectroscopy (Renishaw, RENISHAW pic, New Mills, UK) in the range of $100-1800$ cm$^{-1}$ was used for offline analysis of the polymorphic fraction of glycine in the solid products. ATR-FTIR (ReactIR 15, DiComp probe with AgX halide fiber, Mettler-Toledo, Columbia, MD, USA) was used to monitor glycine concentration in the solution. For analysis of glycine concentration, the calibration method suggested by Togkalidou et al. was adopted [31]. The solution was scanned 128 times in the region of 650–4000 cm$^{-1}$ with a resolution of 4 cm$^{-1}$. Here, the spectrum of pure water was used as the background, and spectral data were acquired using iC IR software (Ver. 7.0, Mettler-Toledo, Columbia, MD, USA) for ATR-FTIR analysis.

### 3. Result and Discussion

The α-form and γ-form polymorphs of glycine were identified by PXRD patterns, as shown in Figure 2. The PXRD patterns for α-form and γ-form crystals differed due to the different molecular arrangements in their crystalline structures. The α-form structure was identified by characteristic peaks at 2θ of ~19.05°, ~29.9°, and ~35.4°, and the γ-form structure was identified by characteristic peaks at 2θ of ~21.8°, ~25.3°, and ~39.1° [30,32]. Here, all the PXRD patterns were obtained with a slow scan rate of 0.02 °/s, and no characteristic peaks of β-form were observed at ~18.0° or ~24.8°, which also confirmed that there was no β-form in the samples [30,32,33]. Moreover, the α-form and γ-form polymorphs of glycine were identified by FT-Raman spectroscopy, as shown in

Figure 3. The FT-Raman spectra for mixtures of $\alpha$-form and $\gamma$-form polymorphs at various mass fractions are presented in Figure 3a. The unique absorption peak areas for $\gamma$-form and $\alpha$-form were detected at 1325–1360 cm$^{-1}$ and 1445–1465 cm$^{-1}$, respectively. As such, when decreasing the faction of $\gamma$-form in the polymorph mixture, the peak area at 1325–1360 cm$^{-1}$ was reduced, and the peak area at 1325–1360 cm$^{-1}$ disappeared at 0 wt% of $\gamma$-form. The highest peak area was displayed at 1445–1465 cm$^{-1}$ for 0 wt% of $\gamma$-form (i.e., 100 wt% of $\alpha$-form); then, the peak area at 1445–1465 cm$^{-1}$ was reduced as the fraction of $\alpha$-form was decreased. Based on these characteristic peak areas of two forms from FT-Raman spectroscopy, the fraction of $\gamma$-form glycine in a polymorphic mixture of $\gamma$-form and $\alpha$-form was calibrated using Equation (1).

$$\text{Area ratio} = \left(\text{Area}_{1325-1360\ cm^{-1}}\right) / \left(\text{Area}_{1325-1360\ cm^{-1}} + \text{Area}_{1445-1465 cm^{-1}}\right) \qquad (1)$$

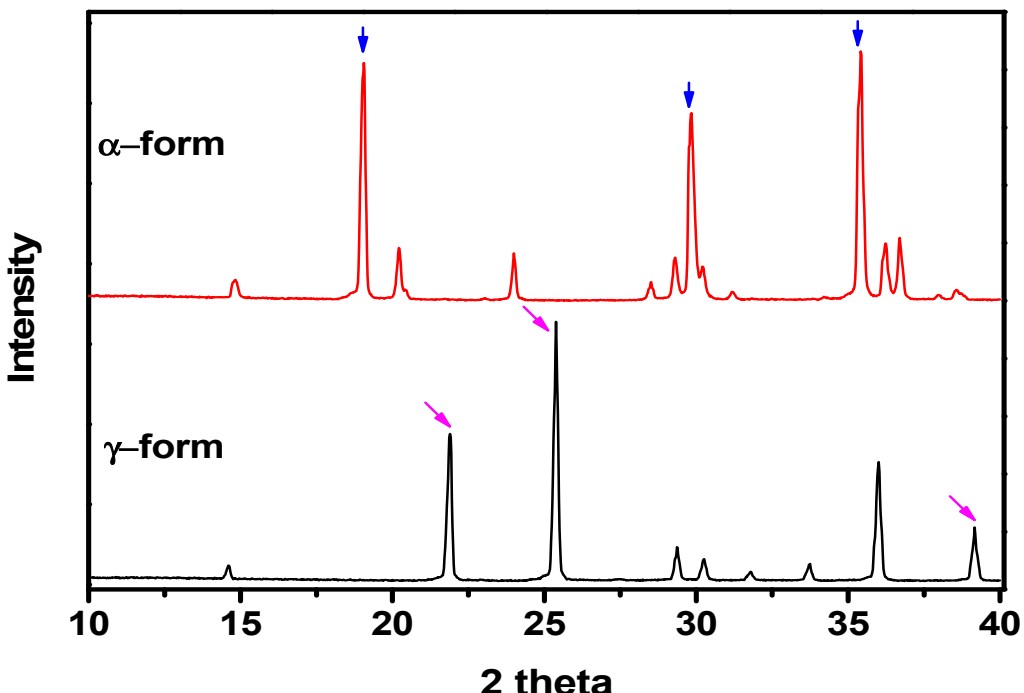

**Figure 2.** PXRD patterns of the $\alpha$-form and $\gamma$-form glycine crystals. The bule arrows indicate the characteristic peaks of $\alpha$-form (~19.05°, ~29.9°, and ~35.4°), and the pink arrows indicate the characteristic peaks of $\gamma$-form (~21.8°, ~25.3°, and ~39.1°).

The result for standard calibration of peak area ratio and polymorphic fraction is plotted in Figure 3b. The linear fitting equation with $R^2$ = 0.9898 was applied to estimate the polymorphic fraction in crystal samples during phase transformation. Here, the X axis is the mass fraction of $\gamma$-form glycine, and the Y axis is the peak area ratio.

The influence of agitation speeds on the polymorphic phase transformation of glycine $\alpha$-form to $\gamma$-form is shown in Figure 4a. Phase transformation behavior was monitored in the present study by changes in the $\gamma$-form and $\alpha$-form fractions. Because the pure $\alpha$-form crystals were always used as seeds to study the phase transformation from $\alpha$-form to $\gamma$-form, the initial fraction of $\gamma$-form was always 0 wt%. At an agitation speed of 750 rpm, the $\gamma$-form fraction maintained 0 wt% within the initial 25 h; then, it became slightly larger than 0 wt% after 25 h, indicating the induction of nucleation of stable $\gamma$-form. Because the solution at this time was supersaturated with respect to $\gamma$-form, the subsequent growth of $\gamma$-form caused its fraction in the suspension to gradually increase with time. Simultaneously, the nucleation and growth of the $\gamma$-form consumed the solute in the solution that was initially saturated with respect to the $\alpha$-form, further promoting the dissolution of the $\alpha$-form. As such, the $\gamma$-form fraction eventually reached 100 wt% as a result of a complete

transition of α-form to γ-form after 55 h. However, when increasing the agitation speed to 1200 rpm, the time for induction of nucleation of γ-form was reduced to 12 h. Additionally, the complete conversion of α-form to γ-form was significantly reduced to 29 h when compared to 55 h at an agitation speed of 750 rpm. Here, the promotion of the nucleation of stable γ-form and the transformation of metastable α-form to γ-form was attributed to the enhanced molecular motion and mass transfer under high agitation speed. As reported by Liu et al., the induction time in polymorphic crystallization decreased with increasing agitation speed [11].

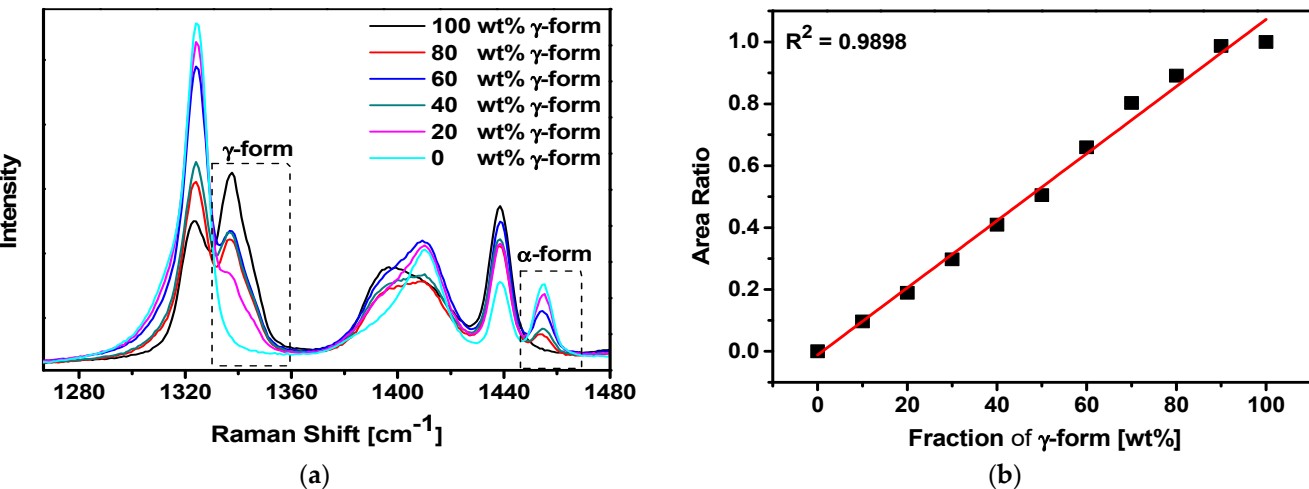

**Figure 3.** Analysis of the polymorphic fraction of glycine using FT-Raman spectroscopy: (**a**) comparison of characteristic relative peak intensities for the polymorphic mixture of α-form and γ-form at various γ-form weight fractions; (**b**) standard calibration of γ-form characteristic peak area ratio and γ-form weight fraction in the polymorphic mixture of α-form and γ-form.

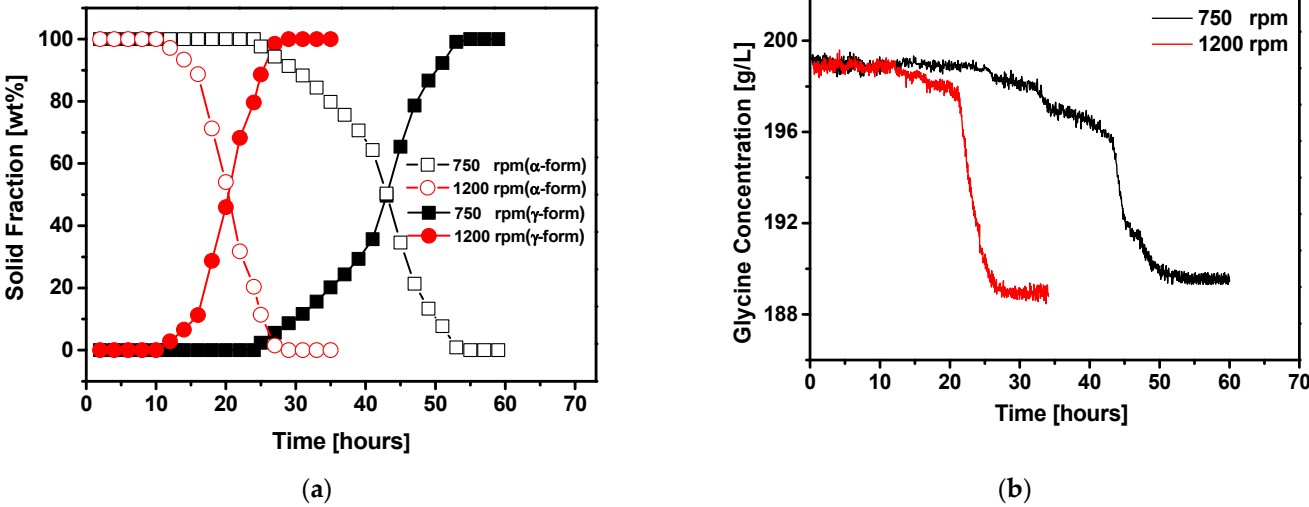

**Figure 4.** Influence of agitation speed on the polymorphic phase transformation of glycine α-form to γ-form. (**a**) Solid fraction of γ-form and α-form glycine; (**b**) glycine solution concentration during crystallization. The crystallization temperature, NaCl concentration, and seed concentration were fixed at 20 °C, 80 g/L, and 60 g/L, respectively.

The glycine concentration in solution during phase transformation was monitored using in situ ATR-FTIR, as shown in Figure 4b. Initially, the glycine concentration was maintained at approximately 199 g/L, which corresponds to the saturation concentration of α-form. Then, after approximately 25 h, the solution concentration at 750 rpm began to decrease and remained unchanged after it dropped to approximately 189 g/L. The time

at which the concentration of the glycine solution began to decrease coincides well with the time for nucleation of stable γ-form. The time at which the solution concentration reached a constant value again corresponds to a complete phase conversion from α-form to γ-form, i.e., approximately 30 h. Because the nucleation of γ-form is promoted at a higher agitation speed of 1200 rpm, the solution concentration started to decrease from 12 h and dropped to approximately 189 g/L within approximately 18 h. Although the initial and final solution concentrations were consistent at various agitation speeds, the time consumed was significantly reduced from 30 h to 18 h with increased agitation speed. This means that increasing the agitation speed not only promoted nucleation of stable γ-form but also accelerated the crystal dissolution and growth process.

As the solution concentration is reduced, the speed reduction in the early stage is slow, whereas the speed reduction in the later stage increases sharply. The slow decline of the solution concentration in the early stage may be related to the low fraction of solid γ-form in the solution. When the fraction of γ-form is very low, only a small amount of solute can be consumed by limited crystal growth. In addition, the fraction of α-form is still high at this time, and its dissolution can continuously transfer solutes into the solution, so the change in solution concentration changing is minimal. With the progress of the phase transition, the fraction of γ-form continuously increases. When the γ-form fraction increases to above 50 wt%, the enhanced growth of γ-form crystals leads to rapid solute consumption, that is, a sharp drop in solution concentration. These behaviors during polymorphic phase transformation are represented in Figure 4. It also suggested that the rate of glycine polymorphic transformation strongly depends on crystal growth of stable γ-form.

The influence of crystallization temperature on glycine polymorphic phase transformation is summarized in Figure 5a. Both the induction time of γ-form and the transformation rate of α-form to γ-form were found to vary significantly with the crystallization temperature. At a crystallization temperature of 20 °C, the induction time of γ-form was about 25 h, and the complete conversion to γ-form from α-form was achieved in 55 h. At a higher crystallization temperature of 30 °C, the γ-form was induced in a shorter time of 11 h, and the transformation was completed within 18 h. When the crystallization temperature was further increased to 40 °C, the induction time of γ-form was significantly reduced to less than 3.5 h; thus, 100 wt% of γ-form was achieved within 7 h. Here, the decrease in γ-form induction time with increasing temperature can be explained in terms of enhanced molecular motion and reduced interfacial energy between the solid and liquid phases. As shown in Figure 5b, the changes in glycine solution concentration were also detected by in situ ATR-FTIR. At a crystallization temperature of 20 °C, the glycine concentration was initially maintained at approximately 199 g/L until 25 h, at which point the concentration began to decrease before eventually reaching approximately 189 g/L after 30 h. When the crystallization temperature was 30 °C, the glycine concentration was maintained at about 246 g/L for 11 h, at which point it decreased to about 238 g/L in 7 h. At a crystallization temperature of 40 °C, the glycine concentration was initially maintained at 279 g/L for about 3.5 h before rapidly decreasing to about 268 g/L in less than 4 h. We found that the time at which the glycine concentration began to decrease was always consistent with the time at which the stable γ-form appeared. This means that the nucleation of a stable γ-form is a critical step in initiating phase transformation. These results also confirm the considerable influence of crystallization temperature on the polymorphic transformation between glycine crystals.

The influence of NaCl as an additive on glycine polymorphic phase transformation was investigated by varying the NaCl concentration in solution, as shown in Figure 6a. At an NaCl concentration of 80 g/L, the time required to reach pure γ-form was about 55 h. The phase transformation of α-form to γ-form was accelerated at a higher NaCl concentration of 120 g/L. As such, the pure γ-form was achieved within 47 h. We also found that the induction time of γ-form was reduced from 25 h to 20 h when the NaCl concentration was increased from 80 g/L to 120 g/L. Here, the nucleation of stable γ-form was facilitated at a

higher NaCl concentration because NaCl in solution can change the packing arrangement of glycine molecules and thereby promote the nucleation of γ-form crystals. As reported by Yang et al., at high concentrations of NaCl, polymorphic crystallization of glycine is biased towards the formation of γ-form [18]. Moreover, because the arrangement of glycine molecules and clusters at high NaCl concentrations can promote nucleation of γ-form, it can also facilitate the crystal growth of γ-form. Therefore, the phase transformation rate was enhanced.

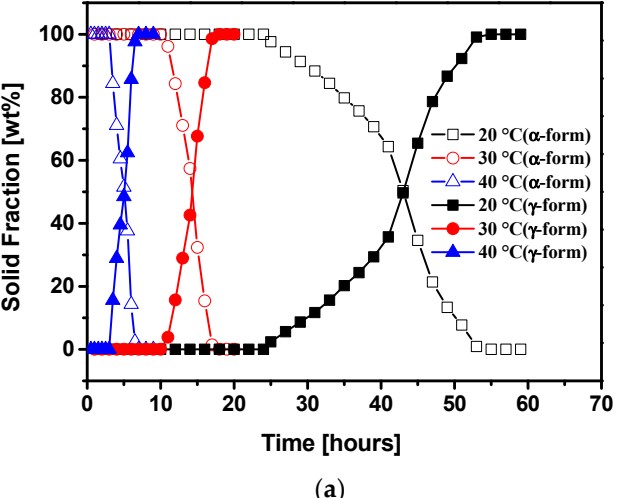
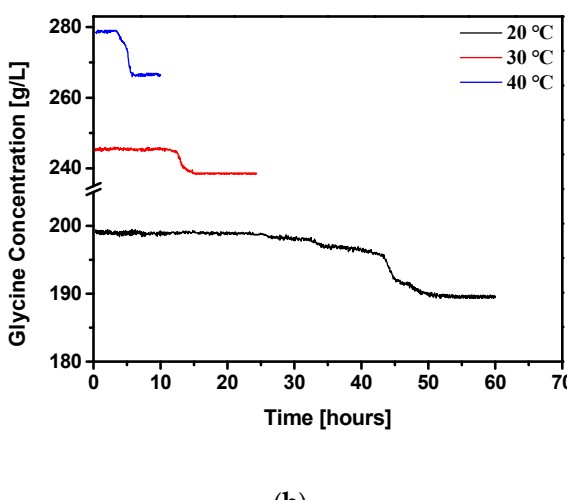

(**a**)
(**b**)

**Figure 5.** Influence of crystallization temperature on glycine polymorphic phase transformation of glycine α-form to γ-form. (**a**) Solid fraction of γ-form and α-form glycine; (**b**) glycine solution concentration during crystallization. The agitation speed, NaCl concentration, and seed concentration were fixed at 750 rpm, 80 g/L, and 60 g/L, respectively.

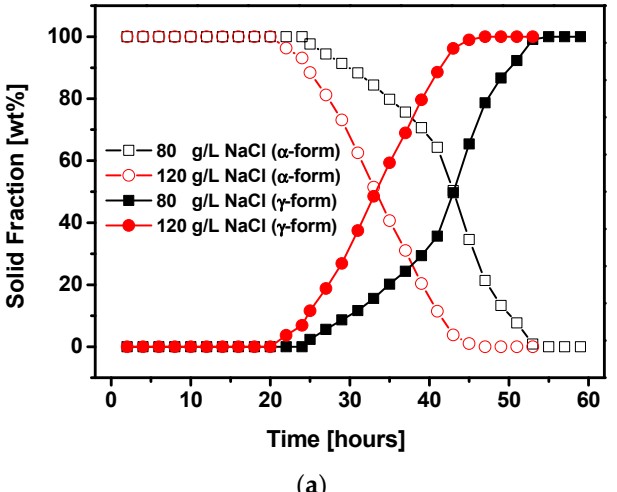
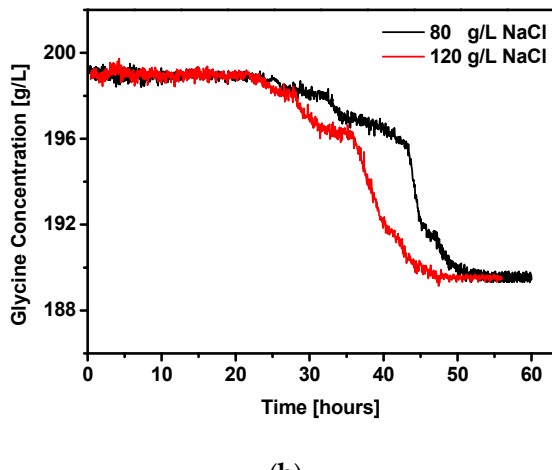

(**a**)
(**b**)

**Figure 6.** Influence of NaCl concentration on glycine polymorphic phase transformation of glycine α-form to γ-form. (**a**) Solid fraction of γ-form and α-form glycine; (**b**) glycine solution concentration during crystallization. The agitation speed, temperature, and seed concentration were fixed at 750 rpm, 20 °C, and 60 g/L, respectively.

The change in glycine solution concentration during phase transformation at various NaCl concentrations is displayed in Figure 6b. The glycine concentration decreased earlier and faster with 120 g/L NaCl compared to 80 g/L NaCl due to the enhanced nucleation and growth of γ-form.

We also found that the concentration of α-form seeds strongly affects the polymorphic phase transformation of glycine. The polymorphic transformation behavior at 60 g/L and

120 g/L $\alpha$-form seed is summarized in Figure 7a. Interestingly, the induction time for stable $\gamma$-form decreased obviously with increased $\alpha$-seed concentration, and the transformation rate was also improved at higher $\alpha$-seed concentration. Thus, at 120 g/L $\alpha$-form seed concentration, the conversion of $\alpha$-form to $\gamma$-form was completed in 31 h, whereas at 60 g/L $\alpha$-form seed concentration, it took 55 h. With increased $\alpha$-form seed loading, the total polymorph crystal surface was increased, which may have enhanced the possibility of nucleation of stable $\gamma$-form. As reported by Croker et al., the nucleation of stable form crystals during polymorphic transformation is likely to occur on the crystal surface of the existing metastable form [34], as the formation of stable $\gamma$-form is a critical step in the initiation of phase transformation. Therefore, it is reasonable that a higher seed concentration would induce faster transformation. This effect of seed concentration on polymorphic transformation was also reflected in changes in the solute concentration of glycine, as shown in Figure 7b. The glycine solution concentration decreased earlier at a higher seed concentration of $\alpha$-form due to the enhanced nucleation of stable $\gamma$-form. It is also obvious that the decrease in the glycine concentration was very small at the beginning of the formation of stable $\gamma$-form in both cases. When the proportion of $\gamma$-form increased to more than 50 wt%, the glycine concentration began to decrease sharply until reaching the saturation concentration of $\gamma$-form due to the considerable growth of $\gamma$-form and complete dissolution of $\alpha$-form. Therefore, higher seed loading also resulted in a faster decrease in glycine solution concentration.

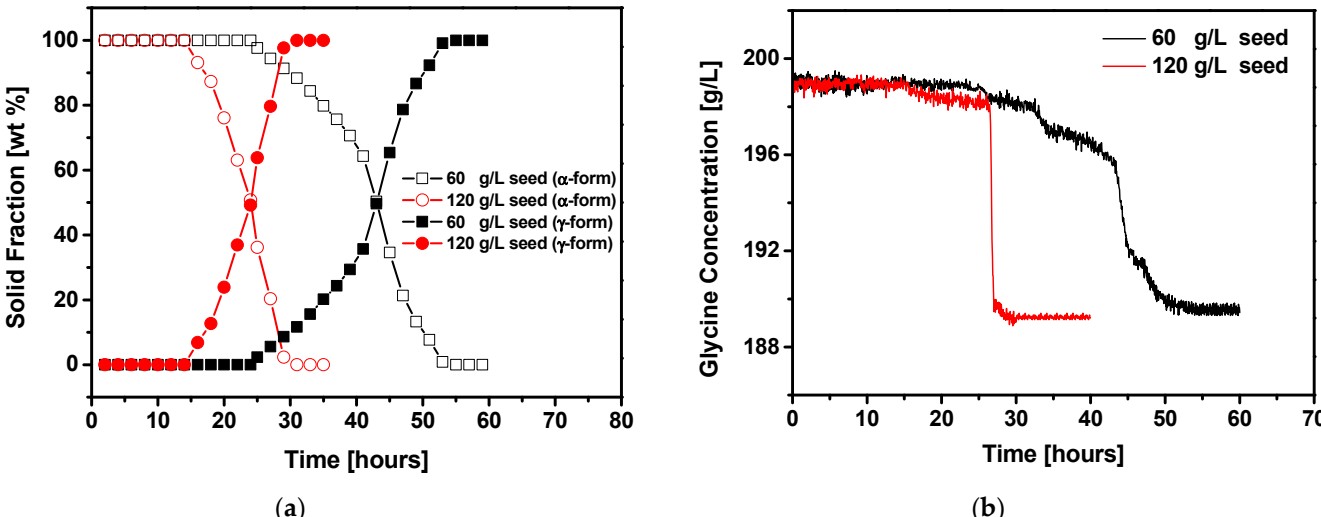

**Figure 7.** Influence of seed concentrations of $\alpha$-form on glycine polymorphic phase transformation of $\alpha$-form to $\gamma$-form. (**a**) Solid fraction of $\gamma$-form and $\alpha$-form glycine; (**b**) glycine solution concentration during crystallization. The agitation speed, temperature, and NaCl concentration were fixed at 750 rpm, 20 °C, and 80 g/L, respectively.

## 4. Conclusions

We found that the polymorphic phase transformation of glycine was strongly influenced by several process parameters, including agitation speed, crystallization temperature, and the seed concentration of metastable $\alpha$-form. Furthermore, the concentration of NaCl as an additive was also an important influencing factor with respect to polymorphic transformation between glycine crystals. As such, the phase transformation of metastable $\alpha$-form to stable $\gamma$-form was accelerated by increasing the agitation speed and the crystallization temperature. The transformation rate increased with increased seed concentration of $\alpha$-form. Additionally, the increase in NaCl concentration in aqueous solution facilitated phase transformation of glycine due to the promotion of nucleation and growth of $\gamma$-form. We also identified that the key step for initiating phase transformation is the induction of stable $\gamma$-form nucleation. Furthermore, the growth of $\gamma$-form is crucial for determining the rate of phase transformation. These behaviors of glycine polymorphic phase transitions were

monitored by analyzing the solid composition and solution concentration changes using FT-Raman spectroscopy and in situ ATR-FTIR, respectively. This study not only provides practical and effective insights for improving the phase transformation process but also provides reliable monitoring and characterization methods for the study of polymorphic transformations, which has important implications for both industrial production and academic research.

**Author Contributions:** Data curation, investigation, visualization, and writing—original draft, B.Z.; conceptualization, supervision, and writing—review and editing, Z.L. All authors have read and agreed to the published version of the manuscript.

**Funding:** This work is funded by the Scientific Project of Hunan Provincial Department of Education (No. 20B253) and the Scientific Research Project of Hunan University of Science and Technology (No. 20XKY067).

**Institutional Review Board Statement:** Not Applicable.

**Informed Consent Statement:** Not applicable.

**Data Availability Statement:** Not Applicable.

**Conflicts of Interest:** The authors declare no conflict of interest.

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
