# Peer review of "Investigation of Glycine Polymorphic Transformation by In Situ ATR-FTIR and FT-Raman Spectroscopy"

_crystals, doi:10.3390/cryst12081141_

Round 1

Reviewer 1 Report

The manuscript entitled: Investigation of Glycine Polymorphic Transformation by in- 2 situ ATR-FTIR and FT-Raman Spectroscopy, by Zhang and Li describes a systematic study on the stabilization of gamma glycine while crystal growth process. Different factors of the growth are changed separately (temperature, concentration, stirring) while growth evolution. PXRD and ATR-FTIR are used for characterization, the analysis is performed in a professional way. The results of the experiments are coherent and explained very well. Novelty is still good enough in the context authors present their work. I think that the manuscript is worth publishing in Crystals after a minor revision, as it satisfies the scientific quality, and it appeals to a wide readership of the journal. The following minor comments must be addressed by the authors prior to publication:

Authors must define the experimental terms when they first appear in the text, such as ATR-FTIT, PXRD.

In principle, the beta polymorph might grow in specific growth conditions ( https://doi.org/10.1007/s10853-018-03211-y ) . In the manuscript, authors do not relate to this polymorph. Even more, the XRD peaks of beta and alpha polymorphs are close to each other (see same reference). Did the authors perform the XRD scan with enough small step size to distinguish between the phases? Though the alpha phase is confirmed by FTIR, conclusions should not be made based on XRD.

Terminology: the first paragraph of Results and Discussion, authors talk about intensity of peaks, but in equation (1) authors relate to the area of a peak. Please be precise.

The link of consumed amount of Glycine to the polymorph growth is indeed nice, it correlates with ( same reference from before, and other references) where the phase is determined by availability of the glycine in solution. This should be addressed.

Do authors recognize any preferred crystallographic orientation of the growth glycine? Or do the peaks relative intensity match the powder diffraction?

Author Response

Overall comment: The manuscript entitled: Investigation of Glycine Polymorphic Transformation by in- 2 situ ATR-FTIR and FT-Raman Spectroscopy, by Zhang and Li describes a systematic study on the stabilization of gamma glycine while crystal growth process. Different factors of the growth are changed separately (temperature, concentration, stirring) while growth evolution. PXRD and ATR-FTIR are used for characterization, the analysis is performed in a professional way. The results of the experiments are coherent and explained very well. Novelty is still good enough in the context authors present their work. I think that the manuscript is worth publishing in Crystals after a minor revision, as it satisfies the scientific quality, and it appeals to a wide readership of the journal. The following minor comments must be addressed by the authors prior to publication:

Response: we would like to thank the reviewer’s positive comment and have revised the manuscript accordingly.

Comment 1: Authors must define the experimental terms when they first appear in the text, such as ATR-FTIR, PXRD.

Response: Thank you for your comment, we have defined terms of ATR-FTIR, FT-Raman and PXRD when they first appeared in the text. Please see lines 12, 13 and 108 in the revised manuscript.

Comment 2: In principle, the beta polymorph might grow in specific growth conditions (https://doi.org/10.1007/s10853-018-03211-y). In the manuscript, authors do not relate to this polymorph. Even more, the XRD peaks of beta and alpha polymorphs are close to each other (see same reference). Did the authors perform the XRD scan with enough small step size to distinguish between the phases? Though the alpha phase is confirmed by FTIR, conclusions should not be made based on XRD.

Response: Many thanks to the reviewer’s important comment. All the PXRD patterns in the present work were obtained under very slow scan rate of 0.02 °/s to distinguish the structure of different phases. According to literatures, the α-form structure was confirmed by the characteristic peaks at 2θ of ~19.05°, ~29.9°, and ~35.4°, and the γ-form structure was confirmed by the characteristic peaks at 2θ of ~21.8°, ~25.3°, and ~39.1°. In addition, no characteristic peaks of β-form were observed at ~18° and ~24.8°, which also confirmed that there was no β-form in the samples. In response to your comment, we have included the above description into the manuscript. Please see lines 124-129 in the revised manuscript.

Comment 3: Terminology: the first paragraph of Results and Discussion, authors talk about intensity of peaks, but in equation (1) authors relate to the area of a peak. Please be precise.

Response: Thank you for your comment, we have modified the “peak intensity” uniformly to the “peak area”. Please see lines 132-140 in the revised manuscript.

Comment 4: The link of consumed amount of Glycine to the polymorph growth is indeed nice, it correlates with (same reference from before, and other references) where the phase is determined by availability of the glycine in solution. This should be addressed.

Response: Thanks to the reviewer for her/his important comment. In the present study, we confirmed that phase transformation was proceeded from α-form to γ-form, please refer to responses in comment 2. Actually, the experiments of this study were always started from saturated aqueous solution of metastable α-form with the existence of sufficient amount of α-seed. Therefore, the crystallization of both α-glycine and unstable β-glycine during isothermal phase transitions was not supported kinetically and thermodynamically. In contrast, the glycine in solution was available for crystallization of stable γ-form, as the initial glycine concentration was supersaturated with respect to less soluble γ-form.

Comment 5: Do authors recognize any preferred crystallographic orientation of the growth glycine? Or do the peaks relative intensity match the powder diffraction?

Response: Thanks for your valuable comment. For γ-form glycine, the peaks relative intensity well matched the powder X-ray diffraction, so we think there was no preferred crystallographic orientation. In the case of α-form, a higher peak intensity was observed at the position of ~35.4°, corresponding to the (041) crystallographic plane. However, since we did not pay too much attention to this issue during the experiment before, it has not been discussed in the present version of manuscript. We will pay more attention to these phenomena in future research.

Reviewer 2 Report

The submitted manuscript is of high quality, in terms of both novelty and presentation style. It describes the application of combined IR/Raman spectroscopies to study the polymorphic phase transition through dissolution and subsequent recrystallization. While I think this work deserves to be published, it also requires some revision.

Line 37, I can’t fully agree with this statement. On the one hand more stable form is indeed safer option as we will not observe the phase transition during the storage of a drug. But on the other hand the most stable form is usually the one with the lowest solubility, which is crucial for maintain high bioavailability. For example, the famous ritonavir case. I suggest rewriting this part.

In the introduction the Authors must describe the polymorphism of glycine, the studied compound. It is very complex and include multiple forms, some of them are stable only at elevated pressure. However, apart from alpha and gamma forms, there is also beta form that can be found at normal conditions, however it is unstable. It is therefore necessary to confirm that the beta form was not obtained during your experiment and, what you have observed, was indeed alpha to gamma direct transition. It is important as the background of the study.

I think that the Authors should correlate (compare) the results from Figures 4-7 “a” and “b”. Since the equilibrium solubility of each polymorph is known and there are only two of them (at least this is what the Authors assume) they should have calculate the “fraction of each form” by assuming that the glycine concentration (known, presented in “b”) is the sum of (equilibrium dissolution, which is known) multiplied by (fraction, which is unknown) for the two polymorphs. In that way the Authors should receive the results similar to those presented in the “a” (look at the equation below):

GC=EDA*FA + EDG*(1-FA) where GC is glycine concentration (from “b), EDA is dissolution of alpha at those conditions and EDG is dissolution of gamma at those conditions

Line 107, PXRD is not a spectroscopic method.

Line 108 and further, there is no such thing as “PXRD spectra”, only “PXRD patterns”

Line 157, was it 700 or 750 rpm?

At the end, the part describing each Author’s individual contribution is missing. This is mandatory in Crystals.

Author Response

Overall comment: The submitted manuscript is of high quality, in terms of both novelty and presentation style. It describes the application of combined IR/Raman spectroscopies to study the polymorphic phase transition through dissolution and subsequent recrystallization. While I think this work deserves to be published, it also requires some revision.

Response: we would like to thank the reviewer’s positive comment and valuable suggestions. In response to your comment, the manuscript has been carefully revised.

Comment 1: Line 37, I can’t fully agree with this statement. On the one hand more stable form is indeed safer option as we will not observe the phase transition during the storage of a drug. But on the other hand the most stable form is usually the one with the lowest solubility, which is crucial for maintain high bioavailability. For example, the famous ritonavir case. I suggest rewriting this part.

Response: We are very grateful to the reviewer for the valuable suggestion. Per your comment, we have revised this introduction part as below:

“According to Ostwald's rule of stages, polymorphic crystallization is theoretically easier to first form a metastable form with higher free energy because the energy barrier to over-come is lower. Afterwards, the metastable phase can undergo phase transformation to form more stable form. The stable form is often favored when considering the safety and stability of the drug during storage. However, it is worth noting that usually stable form has the lowest solubility, which may limit the bioavailability of drugs. A typical example is the antiviral drug ritonavir, the metastable form I of ritonavir used to treat Acquired immunodeficiency syndrome (AIDS). However, when form I was converted into the more stable form II, the solubility was greatly reduced, so that the therapeutic effect cannot be achieved. Based on the above facts, the polymorphic phase transformation during polymorph crystallization has always received extensive attention”.

Please see lines 32-43 in the revised manuscript.

Comment 2: In the introduction the Authors must describe the polymorphism of glycine, the studied compound. It is very complex and include multiple forms, some of them are stable only at elevated pressure. However, apart from alpha and gamma forms, there is also beta form that can be found at normal conditions, however it is unstable. It is therefore necessary to confirm that the beta form was not obtained during your experiment and, what you have observed, was indeed alpha to gamma direct transition. It is important as the background of the study.

Response: Many thanks to the reviewer’s important and valuable comment. In response to your comment, we have revised the Introduction section and added a description of glycine and glycine polymorphs. Please see lines 66-74 in the revised manuscript.

In addition, we performed all the PXRD analysis with a very slow scan rate of 0.02 °/s to distinguish the structures of different phases. Based on the literature, the α-form structure was confirmed by the characteristic peaks at 2θ of ~19.05°, ~29.9°, and ~35.4°, and the γ-form structure was confirmed by the characteristic peaks at 2θ of ~21.8°, ~25.3°, and ~39.1°. Also, no characteristic peaks of β-form were observed at ~18° and ~24.8°, which confirmed that there was no β-form in the samples. In response to your comment, we have included the above description into the manuscript. Please see lines 124-129 in the revised manuscript.

Comment 3: I think that the Authors should correlate (compare) the results from Figures 4-7 “a” and “b”. Since the equilibrium solubility of each polymorph is known and there are only two of them (at least this is what the Authors assume) they should have calculate the “fraction of each form” by assuming that the glycine concentration (known, presented in “b”) is the sum of (equilibrium dissolution, which is known) multiplied by (fraction, which is unknown) for the two polymorphs. In that way the Authors should receive the results similar to those presented in the “a” (look at the equation below):

GC=EDA*FA + EDG*(1-FA) where GC is glycine concentration (from “b), EDA is dissolution of alpha at those conditions and EDG is dissolution of gamma at those conditions

Response: Per your comment, solid fractions of both γ- and α-form glycine during phase transformation were plotted in Figures 4-7. Please see figures 4a, 5a, 6a and 7a in the revised manuscript.

Comment 4:

Line 107, PXRD is not a spectroscopic method.

Line 108 and further, there is no such thing as “PXRD spectra”, only “PXRD patterns”

Response: Per your comment, we have corrected the “PXRD spectra” to “PXRD patterns” in the manuscript. Please see lines 122, 123 and 127 in the revised manuscript.

Comment 5: Line 157, was it 700 or 750 rpm?

Response: Thanks for your comment, it should be 750rpm, we have corrected the “700rpm” to “750rpm”. Please see line 176 in the revised manuscript.

Comment 6: At the end, the part describing each Author’s individual contribution is missing. This is mandatory in Crystals.

Response: Thanks for your comment, we have already added the author’s individual contribution in the revised manuscript, please see line 301-303.

Round 2

Reviewer 2 Report

The Authors have corrected their manuscript accordingly. Current version can be accepted for publication.